# Cultivation Factors That Affect Amyloid-β Aggregation Inhibitory Activity in *Perilla frutescens* var. *crispa*

**DOI:** 10.3390/foods12030486

**Published:** 2023-01-20

**Authors:** Keiya Shimamori, Tomohiko Nambu, Daiki Kawamata, Masahiro Kuragano, Naoki Nishishita, Toshifumi Iimori, Shinya Yamanaka, Koji Uwai, Kiyotaka Tokuraku

**Affiliations:** 1Graduate School of Engineering, Muroran Institute of Technology, Muroran 050-8585, Japan; 2Regenerative Medicine and Cell Therapy Laboratories, Kaneka Corporation, Kobe 650-0047, Japan

**Keywords:** Alzheimer’s disease, amyloid-β, quantum dot, *Perilla frutescens* var. *crispa*, cultivation method, fertilization

## Abstract

Alzheimer’s disease (AD) is thought to be caused by the deposition of amyloid-β (Aβ) in the brain. Aβ begins to aggregate approximately 20 years before the expression of its symptoms. Previously, we developed a microliter-scale high-throughput screening (MSHTS) system for inhibitors against Aβ aggregation using quantum dot nanoprobes. Using this system, we also found that plants in the *Lamiaceae*, particularly *Perilla frutescens* var. *crispa*, have high activity. The cultivation environment has the potential to enhance Aβ aggregation inhibitory activity in plants by changing their metabolism. Here, we report on cultivation factors that affected the activity of *P. frutescens* var. *crispa* cultivated in three fields under different cultivation conditions. The results revealed that the activity of *P. frutescens* var. *crispa* harvested just before flowering was highest. Interestingly, the activity of wind-shielded plants that were cultivated to prevent exposure to wind, was reduced to 1/5th of plants just before flowering. Furthermore, activity just before flowering increased following appropriate nitrogen fertilization and at least one week of drying from the day before harvest. In addition, we confirmed that the *P. frutescens* var. *crispa* leaf extracts suppressed Aβ-induced toxicity in nerve growth factor-differentiated PC12 cells. In this study, we demonstrated that flowering, wind, soil water content, and soil nitrogen content affected Aβ aggregation inhibitory activity, necessary to suppress Aβ neurotoxicity, in *P. frutescens* var. *crispa* extracts. This study provides practical cultivation methods for *P. frutescens* var. *crispa* with high Aβ aggregation inhibitory activity for the prevention of AD.

## 1. Introduction

Alzheimer’s disease (AD) is the most common form of dementia and an acute medical issue due to the increasing number of AD patients worldwide. It is estimated that the number of people with dementia will increase from 57.4 million worldwide in 2019 to 152.8 million cases in 2050 [1]. Furthermore, the value of a statistical life-based global economic burden of Alzheimer’s disease and related dementias was estimated at USD 2.8 trillion in 2019 and was projected to increase to USD 16.9 trillion in 2050 [2]. AD progressively causes memory loss and cognitive dysfunction, eventually requiring full-time care [3]. Most studies target amyloid-β (Aβ) peptides that mainly consist of 40 or 42 amino residues [4]. Aβ grows into oligomers, fibrils, and aggregates and is observed as plaques in a patient’s brain [5]. Thus, Aβ aggregation is an initiating factor in AD that leads to the death of neurons [6,7]. Therefore, many studies have been performed aimed at decreasing Aβ aggregation [8].

Nevertheless, most clinical trials targeting AD patients failed [9]. These results might stem from Aβ aggregation that begins about two decades before the onset of symptoms [10], suggesting that treatment for patients in whom AD has developed is too late. Recently, a few clinical trials achieved better results using a treatment based on anti-Aβ antibodies in patients with mild cognitive impairment (MCI) and mild AD. Primarily, the US Food and Drug Administration approved aducanumab, which is a therapeutic medicine based on anti-Aβ fibril antibodies, via an accelerated approval pathway [11]. The treatment of patients with MCI and mild AD with aducanumab showed the removal of Aβ in the brain and reduced AD pathology [12]. In addition, treatment with lecanemab, an anti-Aβ protofibril antibody, showed an even greater reduction in cognitive decline in patients with MCI and mild AD [13]. These achievements support the concept that it is important to inhibit Aβ aggregation before the onset of AD symptoms. However, treatments with antibodies are very expensive [14] and are thus not accessible to most people.

Even if Aβ aggregates are removed after the neural network is severely damaged, it is expected that the network will not recover to its initial state due to its irreversibility. We believe that it is important to prevent neural networks from being damaged. In addition to treatment by removing Aβ aggregation with anti-Aβ antibody drugs, we suggest the importance of adding a preventive effect against Aβ aggregation to the daily diet. Since food enables us to take any substance affordably and permanently, we assume that food with Aβ aggregation inhibitory activity contributes greatly to AD prevention. Previously, we developed a microliter-scale high-throughput screening (MSHTS) system to evaluate Aβ_42_ aggregation inhibitors in vitro by using quantum dot (QD) nanoprobes [15]. The MSHTS system enables high-throughput screening and the accurate evaluation of samples, such as foods containing various contaminants [16,17,18,19]. Generally, atomic force microscopy and electron microscopy (EM) have been used to directly observe Aβ fibrils [20]. In particular, the latest cryo-EM provides high-resolution images of the structure of amyloid fibrils [21]. Although these methods are powerful tools for observing the structure of fibrils, they are not suited for quantification and high-throughput screening. Generally, Aβ aggregation is quantified using thioflavin T (ThT) [22], which measures the amount of fluorescence bound to the β-sheet. However, the ThT assay cannot accurately evaluate activity in contaminated samples because fluorescence is inhibited by many dyes in the samples [16]. In contrast, the MSHTS system allows the accurate evaluation of total Aβ inhibitory activity in various samples, such as crude extracts.

Using the MSHTS system, we have already evaluated the Aβ aggregation inhibitory activity of commercial dressings, which are processed foods, and reported those findings in this journal [16]. Furthermore, we applied this system to food plants and found that one family in particular, the *Lamiaceae*, displayed high activity [15]. In addition, we attempted to isolate active compounds from summer savory, a member of the *Lamiaceae*, revealing that the main active compound was rosmarinic acid (RA), a polyphenol. Subsequently, we found that Aβ aggregation inhibitory activity in leaf extracts from *Perilla frutescens* var. *crispa* was higher than from spearmint (*Mentha spicata*), and had the highest activity among all tested members of the *Lamiaceae* [15]. Presumably, this high activity derives from its rich polyphenol content [23,24,25]. However, we preliminarily confirmed that the activity differed, depending on harvest time and cultivation practice, even within *P. frutescens* var. *crispa*. Even in the same plants, the content of polyphenols changed due to the growth cycle and cultivation conditions, such as fertilization [26,27,28], including in perilla [29,30,31].

In this study, we investigated the cultivation factors that affected the Aβ aggregation inhibitory activity of *P. frutescens* var. *crispa*. We evaluate total activity in extracts using the automated MSHTS system, which allows for high-throughput screening [18]. While some papers have indicated cultivation methods for *P. frutescens* var. *crispa* to increase the content of polyphenols and other promising bioactive compounds [28,30], it is unclear whether those contents would contribute to total Aβ aggregation inhibitory activity. Here, we evaluated the activity of *P. frutescens* var. *crispa* cultivated in different fields and fertilization conditions. Consequently, we found that flowering, wind, nitrogen fertilization, and water content affected the activity of *P. frutescens* var. *crispa*. This study provides practical cultivation methods for enhancing Aβ aggregation inhibitory activity.

## 2. Materials and Methods

### 2.1. Materials

Human Aβ_42_ (4349-v, Peptide Institute Inc., Osaka, Japan) and Cys-conjugated Aβ_40_ (23519, Anaspec Inc., Fremont, CA, USA) were purchased commercially. Thiazolyl blue tetrazolium bromide (MTT)(M5655) and poly-D-lysine were purchased from Sigma-Aldrich (St. Louis, MO, USA). Nerve growth factor (NGF) was purchased from Cosmo Bio (Toyo, Japan).

### 2.2. Preparation of QDAβ Nanoprobe

The QDAβ nanoprobe was prepared using QD-PEG-NH_2_ (Qdot^TM^ 605 ITK^TM^ Amino (PEG) Quantum dot; Q21501MP, Waltham, MA, USA, Thermo Fisher Scientific) according to our previous reports [15,18,32]. QDAβ concentration was determined by comparing absorbance at 350 nm to unlabeled QD-PEG-NH_2_.

### 2.3. Evaluation Using the Automated MSHTS System

The half maximal effective concentration (EC_50_) values of all plant extracts were determined by a modified automated MSHTS system, as was described in our previous report [18]. Specifically, mixed solutions were prepared with six concentrations of extracts, 25 nM QDAβ, and 25 μM Aβ_42_ in PBS containing 5% ethanol (EtOH) and 2.5% dimethyl sulfoxide (DMSO) and incubated in a 1536-well plate (782096, Greiner, Kremsmünster, Austria) at 37 °C for 24 h. The images of each well were captured pre- and post-incubation by an inverted fluorescence microscope (see Section 2.4.). Standard deviation (SD) values of the central images of the region of interest (432 × 432 pixels) in each well were measured by the General Analysis program of NIS-Elements (Nikon, Tokyo, Japan). The images on the highest extract concentration, including insoluble substances, were eliminated because these affect SD values and disrupt accurate evaluation. EC_50_ was estimated from the SD values by Prism software (GraphPad software, San Diego, CA, USA) using an EC_50_ shift by global fitting (asymmetric sigmoidal, five-parameter logistic) [15].

### 2.4. Fluorescence Microscopy

Fluorescence images in the 1536-well plate were observed by an inverted fluorescence microscope system (ECLIPSE Ti-E, Nikon) equipped with a color CMOS camera (DS-Ri2, Nikon). QD fluorescence was imaged using a 4× objective lens (MRD00045, Nikon) and a TRITC filter set (TRITC- A-Basic-NTE, Semrock, NY, USA).

### 2.5. MTT Assay

Rat adrenal pheochromocytoma, PC12 cells, were obtained from the JCRB Cell Bank (Osaka, Japan). PC12 cells were maintained in dulbecco’s modified eagle medium supplemented with 10% fetal bovine serum (FBS), 100 U/mL penicillin, and 100 μg/mL streptomycin, as was described in our previous report [33]. Cells were cultured at 37 °C in humidified air containing 5% CO_2_ with CO_2_ incubator. PC12 cells were plated at 1.2 × 10^4^ cells in poly-D-lysine-coated 96-well plates (3860-096, AGC TECHNO GLASS, Shizuoka, Japan) and incubated for 24 h. After incubation, cells were treated with 50 ng/mL NGF for 24 h for neuronal differentiation. After this step, cells were cultured in a medium without FBS but containing 50 ng/mL NGF. Cells were treated with extracts and 25 μM Aβ_42_ for 24 h by replacing the medium with 1% EtOH/1% DMSO. Cells were then treated with 0.5 mM MTT for 4 h by medium replacement in reference to the MTT assay [34]. After incubation, the supernatant of each well was removed, and formazan crystals were dissolved in a 10% SDS/0.01 M HCl solution. After overnight incubation, absorbance (570 nm) was measured in each well with (SH-9000Lab, CORONA ELECTRIC, Ibaraki, Japan). Cell viability was calculated as a percentage relative to cells treated with the control medium (1% EtOH/1% DMSO).

### 2.6. Cultivation of Perilla frutescens *var.* crispa

*P*. *frutescens* var. *crispa* seedlings were purchased from AW Farm Chitose Co., Ltd. (Hokkaido, Japan). Seeds were planted around May and harvested regularly. In 2018, plants were cultivated in Chitose (42°53′24.5″ N 141°41′41.5″ E), Watenbetsu (42°58′08.8″ N 144°00′02.9″ E), and Tomaribetsu (43°01′48.1″ N 144°07′55.2″ E) in Hokkaido, and in 2019, on the field on a college campus (42°22′38.1″ N 141°02′12.1″ E). In 2020, plants were cultivated in a plastic greenhouse on a college campus. In 2018, meteorological data was measured with meteorological equipment installed in each field. In 2019 and 2020, soil from the plot was sampled at the same time as the harvest from the field on the college campus. The water content of the soil was calculated from pre- and post-drying weights.

### 2.7. Preparation of Extracts

Extracts were prepared by extraction with 5 mL of 95% EtOH per gram of fresh leaf weight at room temperature for seven days in 2018. In 2019/2020, perilla leaf extracts were prepared by extraction for one day to prepare many samples efficiently. After the EtOH solution was filtered and concentrated in vacuo, samples were prepared as a 10 mg/mL extract with EtOH.

### 2.8. Statistical Analysis

All statistical analyses were performed in Excel (version 16; Microsoft, Redmond, WA, USA). A two-tailed Welch’s *t*-test was performed for comparisons between groups. Soil water content and EC_50_ were evaluated by linear regression using least squares in Excel. *p* < 0.05 was considered statistically significant (*: *p* < 0.05, **: *p* < 0.01, ***: *p* < 0.005).

## 3. Results

### 3.1. Principle of the MSHTS System

We evaluated more than a hundred samples of *P*. *frutescens* var. *crispa* extracts for Aβ aggregation inhibitory activity using the automated MSHTS System [18] (representative; Figure 1B,C). This system’s uniqueness is in its use of QDs, which are nanoscale crystals with an overwhelmingly non-photobleaching property (Figure 1A). After incubation of a mixture of the fluorescent probe QDAβ and Aβ_42_, QDAβ is incorporated into Aβ_42_ fibrils. From the fluorescence images of the mixtures, the variation in brightness was measured (Figure 1B) as an SD value, which can be considered as the progress of aggregation. Aβ aggregation inhibitory activity was evaluated by developing an inhibition curve using SD values (Figure 1C) and calculating the EC_50_. Lower EC_50_ values indicate higher activity. If the inhibition curve did not reach 50% of the percentage of SD values, then a sample was determined to be not detectable (ND).

### 3.2. Aβ Aggregation Inhibitory Activity of Perilla frutescens *var.* crispa and Meteorological Data in Three Fields

At first, we cultivated *P. frutescens* var. *crispa* plants in three fields in Hokkaido, Japan, in 2018 to investigate regional differences in Aβ aggregation inhibitory activity and installed meteorological equipment in each field in late August (Figure 2). Plants were harvested on three dates (24 and 25 July, 28 and 29 August, and 5 and 10 October). Extracts from leaves, stems, and roots were compared in late July. Leaf extracts showed high activity in all fields, while stem extracts in Tomaribetsu and root extracts in Chitose and Watenbetsu showed no activity (i.e., ND) (Figure 3A). In Chitose and Watenbetsu, roots showed no activity, and leaves showed significantly higher activity than stems. Therefore, we compared the activity of extracts of leaves harvested on the three dates (Figure 3B); the extracts from plants growing in Chitose and Tomaribetsu fields showed higher activity on October 5 and 10 than on July 24 and 25. Furthermore, we compared the activity of leaves harvested on October 5 and 10 in plants growing in different fields: the highest activity was observed in extracts from plants grown in the Chitose field, followed by the Tomaribetsu and Watenbetsu fields.

We also investigated the correlation between meteorological data and Aβ aggregation inhibitory activity. Meteorological data from August 24 to October 4 were acquired from each field and were plotted as a daily average (Figure 4). These data appear to show that low soil water content (Figure 4G) was correlated with the above-mentioned plot-related rank in activity (Chitose < Tomaribetsu < Watenbetsu). Wind speed was highest at Chitose (Figure 4E,F), similar to the activity trend. Thus, we speculated that wind and water content affect Aβ aggregation inhibitory activity.

### 3.3. Aβ Aggregation Inhibitory Activity of Perilla frutescens *var.* crispa Was Highest Just before Flowering and Was Enhanced by the Wind

In 2019/2020, we prepared extracts efficiently with a change in extraction time from seven days to one day and evaluated many samples to investigate the details of cultivation conditions. We confirmed no change of Aβ aggregation inhibitory activity by extraction time (Appendix A). In 2019, plots of various conditions with additional fertilization were made after laying compost on the field in Muroran, and *P*. *frutescens* var. *crispa* was cultivated (Figure 2). Plants cultivated without additional fertilizer were referred to as the standard (Std) and harvested every four weeks. Activity in the leaves of Std plants was highest just before the flowering, on 12 September, decreasing thereafter (Figure 5A). Plants were cultivated in the same soil conditions as Std, but a windbreak was created, enclosing plants with a plastic sheet (Standard with windbreak; Sw)(Figure 5B). In Sw plants, activity just before flowering was significantly lower than that of Std plants, with an EC_50_ that was five-fold higher (Figure 5A). The activity of Sw plants harvested on October 3 was highest, but EC_50_ was still three-fold higher than Std plants harvested on September 5. These results indicate that wind exposure enhanced Aβ aggregation inhibition activity in *P*. *frutescens* var. *crispa*.

### 3.4. Adjustment of Soil Nitrogen and Soil Water Content Enhanced Aβ Aggregation Inhibitory Activity of Perilla frutescens *var.* crispa Just before Flowering

In 2019, *P*. *frutescens* var. *crispa* plants were cultivated under fertilization conditions other than Std, as shown in Table 1. Specifically, the activities of four major types of fertilizers were compared: slaked lime and calcium carbonate powder, scallop shell powder (Sp) and granular (Sg) with zeolite, each superphosphate of lime (SPL), ammonium sulfate (AMS), potassium sulfate (SOP), and a combination of Sp, SPL, AMS, and SOP (Figure 6A–D). There were no significant differences between any treatments in the activity of leaves of plants harvested just before flowering. On the other hand, the activity of plants cultivated on soil enhanced with SPL, AMS, and SOP fertilizer, just before flowering, was highest in AMS-fertilized conditions (Figure 6E).

In 2020, to identify an appropriate amount of AMS fertilizer, *P. frutescens* var. *crispa* plants cultivated with six levels of AMS fertilization were harvested every four weeks (Table 2). Activity in all six fertilization conditions increased until September 9, just before flowering on September 10 (Figure 7A). Just before flowering, Aβ aggregation inhibitory activity and total fresh leaf weight from each plant were the highest in the N3 treatment (Figure 7B,C). In particular, total leaf weight improved from Std to N3 but decreased above N4. These results show that appropriate nitrogen fertilization increased Aβ aggregation inhibitory activity and total leaf.

When *P*. *frutescens* var. *crispa* plants cultivated in all conditions in 2019 were harvested, soil samples were drawn to analyze soil water content (Table 1). We created a scatter plot between soil water content and the activity of leaves and performed linear regression bordering on 20% water content (Figure 8A). The linear regression shows that activity was high at 20% but decreased markedly when water content exceeded 20%. Therefore, to test whether the cessation of watering just before harvest would enhance activity, plants were cultivated in greenhouses in 2020 and watering was stopped every week starting before the day of harvest. Predictably, soil water content decreased when watering was stopped (Figure 8C). On the other hand, Aβ aggregation inhibition activity was enhanced when watering was stopped for one week and higher than on the day before the harvest date. A scatter plot for this data was created, as shown in Figure 8A, although a linear regression was not performed due to too few samples (Figure 8D). The plots with water content exceeding 20% had highest activity. These results indicate that Aβ aggregation inhibitory activity was enhanced by stopping watering before harvest to reduce soil water content.

### 3.5. Suppression Effect of Perilla frutescens *var.* crispa on Aβ-Induced Neurocytotoxicity

We tested whether the *P*. *frutescens* var. *crispa* extract could suppress Aβ-induced neurocytotoxicity by the MTT assay using NGF-differentiated PC12 cells (Figure 9). The data show that 25 μM Aβ_42_ reduced the viability of PC12 cells by 40% relative to the control. The extract significantly restored cell viability and suppressed Aβ-induced neurocytotoxicity the most at 0.781 μg/mL. In contrast, the extracts with more than 25 μg/mL concentrations reduced cell viability to less than 25 μM Aβ and significantly when the concentration was 100 μg/mL.

## 4. Discussion

In this study, we evaluated the Aβ aggregation inhibitory activity of *P*. *frutescens* var. *crispa* cultivated under multiple conditions using the MSHTS system, which has the advantage of directly assessing this activity. At first, we evaluated the activity of plants cultivated in three fields (Figure 3) and found that leaves showed the highest activity among all plant parts that were assessed. In particular, this study’s highest activity of N3 (EC_50_ = 0.0059 mg/mL; Figure 7B) was the second highest in the 504 natural plant extracts previously evaluated [18]. We speculate that this activity is due to polyphenols having Aβ inhibitory effect because perilla leaves are rich in polyphenols [23,24,25]. Several compounds have been reported to show Aβ inhibitory activity among the polyphenols that are found in perilla. In particular, RA is the most promising compound since it is abundant in perilla [23] and has high Aβ aggregation inhibitory activity [15]. Similarly, chalcone [35] and asaron derivatives [36] isolated from perilla have also been reported as having this activity. Since perilla has a wide variety of compounds, total activity is considered by their accumulation and synergistic effect. These secondary metabolites are biosynthesized in response to environmental stress to protect against their oxidation [29,30].

Overall, the Aβ aggregation inhibitory activity of *P*. *frutescens* var. *crispa* increased until early September, until flowering, and then decreased after flowering (Figure 5A and Figure 7A). In perilla, during flowering, the contents of RA and total phenolic compounds become reduced [28,31]. These contents may have been altered by a change in metabolic pathways. The amount of total phenolic compounds in plants decreases with flowering due to the oxidation caused by the action of polyphenol oxidase and peroxidase [37]. Therefore, perilla should be harvested just before flowering to avoid a loss of active compounds.

The Aβ aggregation inhibitory activity showed the same trend as wind speed in *Perilla frutescens* var. *crispa* cultivated in the three fields in 2018 (Figure 3 and Figure 4). Furthermore, *Perilla frutescens* var. *crispa* cultivated without wind exposure showed significantly lower activity (Figure 5). Wind, as an environmental stressor, may promote plant growth and affect secondary metabolites. Wind promotes heat exchange, gas trade, and photosynthesis on plant leaves [38]. It was reported that leaf morphology is altered when plants are exposed to wind, increasing the total phenolics in the leaves [39]. Cultivation with more wind may enhance the activity of plants.

In 2019, Aβ aggregation inhibitory activity increased in soil with less than 20% water content in plants cultivated under multiple fertilization conditions (Figure 8A). Furthermore, watering of plants was ceased just before harvest to induce drought stress, and the cessation of watering for more than one week enhanced the activity (Figure 8C). Phenolic acids increase in response to drought stress in plants [40]. Plants biosynthesize polyphenols and flavonoids to defend themselves from drought [41,42]. Abscisic acid (ABA), a plant hormone, is synthesized in response to drought stress in plants [43]. ABA is involved in the accumulation of phenolic acids, especially anthocyanin, in plants under drought stress [44]. Based on the above, we speculate that drought stress promoted the biosynthesis of compounds that contribute to the Aβ aggregation inhibitory activity in *P*. *frutescens* var. *crispa* via the ABA response pathway.

Aβ aggregation inhibitory activity in *P*. *frutescens* var. *crispa* was highest with appropriate nitrogen fertilization as N3, and lower activity when excessive fertilization was applied (Figure 7B). In general, plants contain high levels of polyphenols when cultivated under low levels of nitrogen fertilization [45]. It may be essential to adjust fertilization according to nutrient availability in the field. Furthermore, the choice of nitrogen supply reduced the total phenolic content of leaves in strawberries, and the influence is compound-specific [46]. Perilla contains not only the main active compounds, such as RA, but also other active compounds [23]. Nitrogen fertilization might alter the composition of compounds to enhance total activity.

We showed that *P*. *frutescens* var. *crispa* leaf extracts suppressed Aβ-induced neurocytotoxicity in neuronally differentiated PC12 cells (Figure 9). The methanolic extracts of *Perilla frutescens* (L.) Britton similarly rescued the Aβ-induced cytotoxicity of PC12 cells [47]. Furthermore, compounds isolated from a *Perilla frutescens* (L.) Britton extract also suppressed Aβ-induced cytotoxicity and exhibited toxicity at high concentrations of 100 μg/mL [36]. In this study, the leaf extract at a high concentration of more than 25 μg/mL reduced cell viability but was lower than Aβ toxicity (Figure 9). Thus, an appropriate concentration of perilla extracts is essential so as not to exhibit toxicity.

This study has a limitation: we did not find other compounds that contribute to the Aβ inhibitory activity of *P*. *frutescens* var. *crispa*. A promising compound seems to be RA because the extracts from fresh perilla leaves prepared similarly in this study contain a high content of RA [48]. Furthermore, we previously reported that RA is the active component of the extract from summer savory (*Satureja hortensis*) in the *Lamiaceae* [15]. Even though we confirmed the presence of RA in the perilla extracts, the content was not enough to assess total activity. Previously, we did not obtain a fraction even equal to total activity, although we attempted to isolate active compounds from *P*. *frutescens* var. *crispa* extracts. This result shows that the activity in the extracts may be due to the synergistic effect of multiple compounds. Total activity is important when assessing the intake of substances such as food. A known example of this synergistic effect is the co-administration of epigallocatechin and gallic acid to improve learning ability in a mouse model of brain senescence [49]. We speculate that multiple compounds show a total activity in *P*. *frutescens* var. *crispa* extracts through synergistic effects and are trying to identify those compounds that contribute to this activity. In addition, it is necessary to research whether these compounds have blood–brain barrier permeability. In a rat model, RA reached the brain via the intraperitoneal administration of extracts from *Plectranthus barbatus* (*Lamiaceae*) herbal tea, including RA [50]. Moreover, perilla leaf extracts improved blocked Aβ aggregates-induced memory impairment and reduced Aβ deposits in the hippocampus with systemic administration in 5XFAD mice with AD-linked mutations [51]. In addition to treatment with anti-Aβ antibody drugs (e.g., aducanumab, lecanemab), prevention with foods such as Perilla may protect many people from AD risk. We expect that this study may contribute to preparing the extracts to show improvement against AD-like pathology.

## 5. Conclusions

In this study, we found several cultivating and harvesting factors of *Perilla frutescens* var. *crispa* that enhanced its Aβ aggregation inhibitory activity. The best harvesting time was just before flowering, and the following cultivation methods were effective: wind, appropriate nitrogen fertilization, and pre-harvest soil drying. These factors may alter the biosynthesis of polyphenols that exhibit this activity, although the active compounds were not identified. Furthermore, we successfully used the MSHTS system to assess total Aβ aggregation inhibitory activity. We consider this method to be a powerful tool for identifying substances with activity in plant-derived food. We hope that the active substances will be discovered and become a preventive and therapeutic agent against AD, for which no cure has yet been found.

## Figures and Tables

**Figure 1 foods-12-00486-f001:**
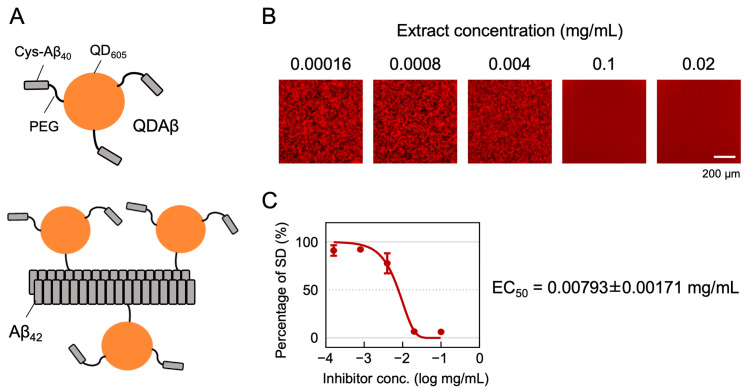
Principle of the MSHTS system for evaluating Aβ_42_ aggregation inhibitory activity. (**A**) QDAβ is a fluorescent probe prepared by cross-linking Cys-Aβ_40_ and QD_605_ via amino polyethylene glycol (PEG)(top). QDAβ co-aggregates with unlabeled Aβ_42_, allowing Aβ_42_ fibrils to be visualized by fluorescence microscopy (bottom). (**B**,**C**) Representative analysis of Aβ aggregation inhibitory activity of *Perilla frutescens* var. *crispa* leaf extract. Mixed solutions of 25 μM Aβ_42_ and extract were added to 1536-well plates and imaged by fluorescence microscopy before and after incubation at 37 °C for 24 h. (**B**) The fluorescence images were trimmed to 432 × 432 pixels, and the SD values, meaning the variation in brightness, were measured. (**C**) The inhibition curve was drawn from SD values, and EC_50_ was calculated using Prism GraphPad software.

**Figure 2 foods-12-00486-f002:**
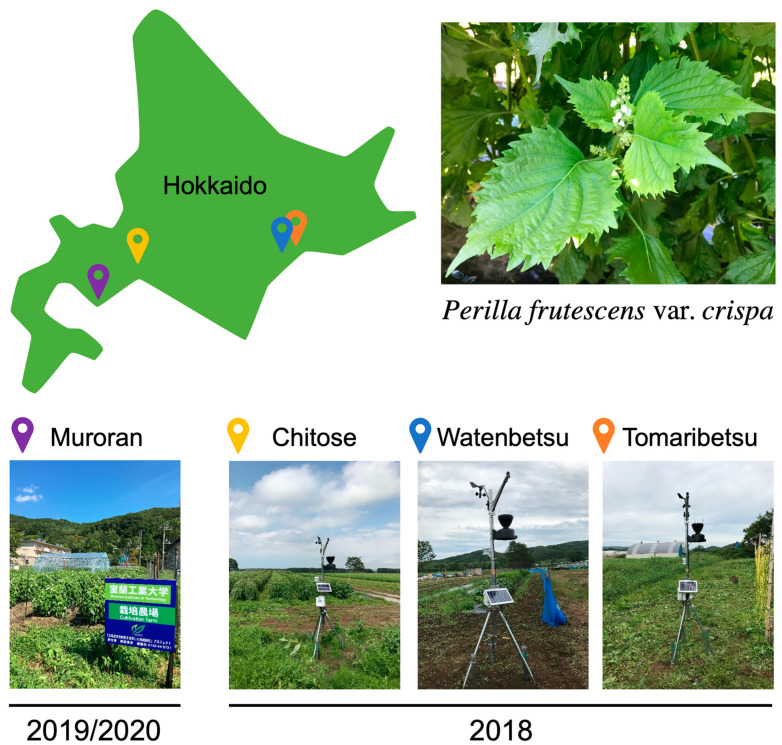
Cultivation of *Perilla frutescens* var. *crispa* in four fields in Hokkaido. In 2018, plants were cultivated in three fields (Chitose, Watenbetsu, and Tomaribetsu) in Hokkaido, Japan, where meteorological equipment was installed. In 2019 and 2020, plants were cultivated under multiple conditions in only Muroran.

**Figure 3 foods-12-00486-f003:**
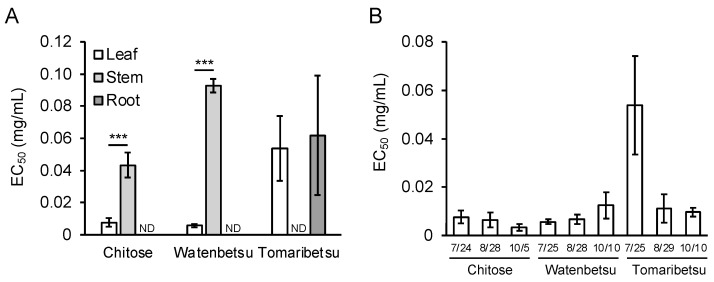
Aβ aggregation inhibitory activity of *Perilla frutescens* var. *crispa* leaves differed depending on fields and harvest time. (**A**) EC_50_ in leaf, stem, and root extracts. Plants were cultivated in three fields in 2018 and harvested in late July. (**B**) The activity of *Perilla frutescens* var. *crispa* leaves was harvested in late July, late August, and early October. The activity was determined to be not detectable (ND) in samples if the inhibitory curve did not reach 50% of the percentage of SD values. Data are represented as mean ± SD (*n* = 4 separate experiments). Comparison of EC_50_ values (***: *p* < 0.005, Welch’s *t*-test).

**Figure 4 foods-12-00486-f004:**
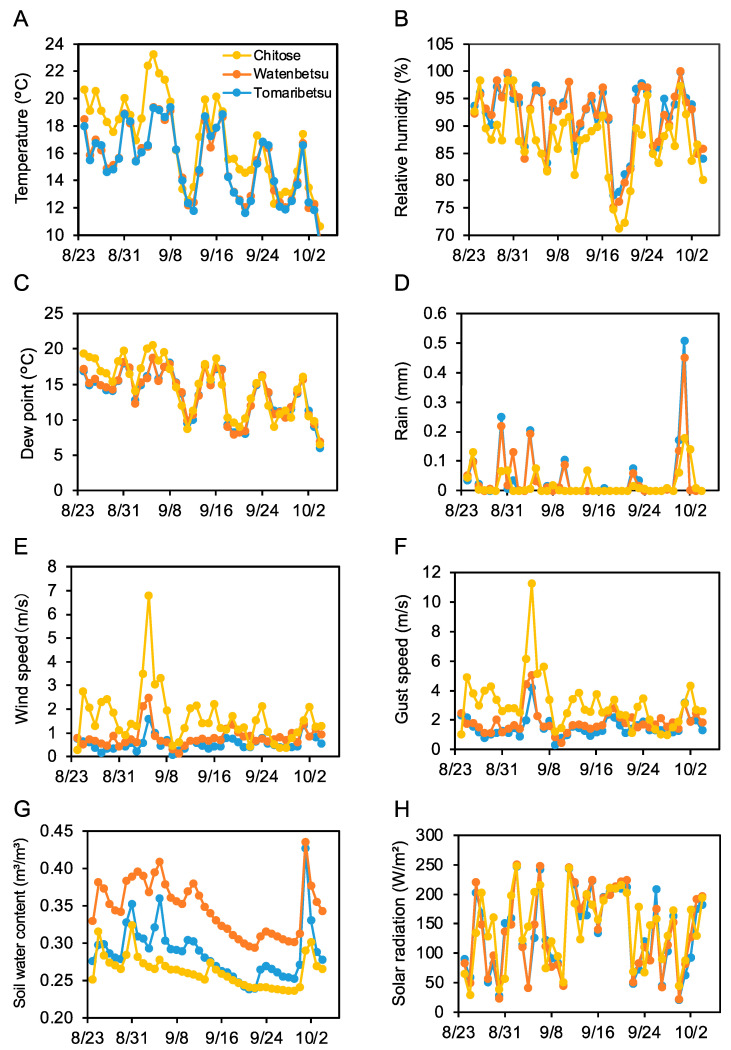
Wind speed and soil water content correlate with regional differences in Aβ aggregation inhibition activity. Meteorological data was measured in three fields from August 24 to October 4, shown as daily averages. Yellow, red, and blue show Chitose, Watenbetsu, and Tomaribetsu, respectively. Temperature (**A**), relative humidity (**B**), dew point (**C**), rain (**D**), wind speed (**E**), gust speed (**F**), soil water content (**G**), and solar radiation (**H**). Data represent an average of each day from every 10 min measurement.

**Figure 5 foods-12-00486-f005:**
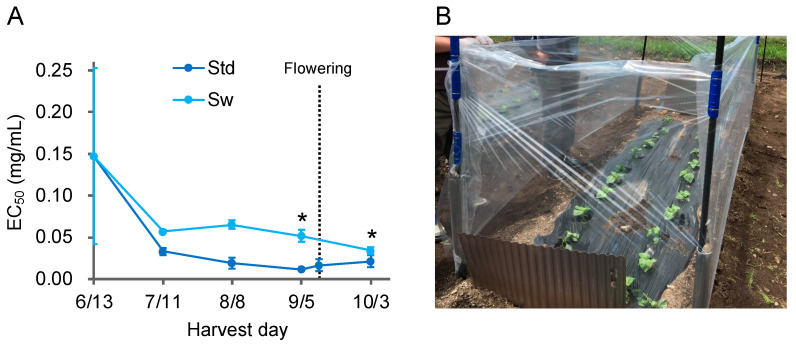
Aβ aggregation inhibitory activity showed the highest just before flowering, with wind enhancement. (**A**) Aβ aggregation inhibitory activity of leaves in regularly harvested *Perilla frutescens* var. *crispa* in 2019. *P*. *frutescens* var. *crispa* was planted on June 13, and whole plants were harvested every 4 weeks. Plants flowered on 12 September. The Std is the condition with no additional fertilizer under all conditions. Plants from Std were also harvested on September 12, during flowering. (**B**) Plants in the standard with windbreak (Sw) were cultivated in the same soil as Std and enclosed with a plastic sheet. Data represent the mean ± SD (*n* = 3, separate plant). Comparison of the EC_50_ values vs. Std harvested on September 5 (*: *p* < 0.05, Welch’s *t*-test).

**Figure 6 foods-12-00486-f006:**
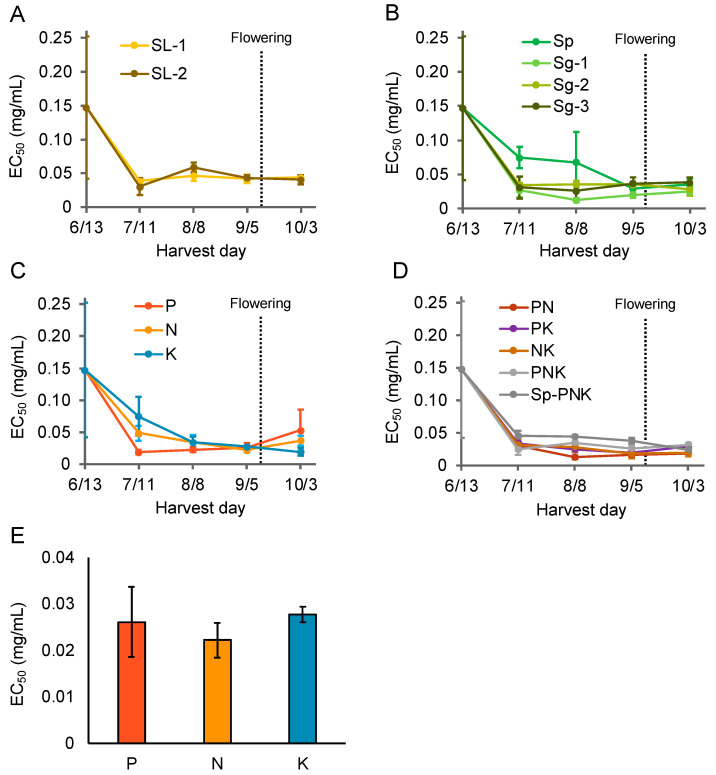
Nitrogen fertilization enhanced the Aβ aggregation inhibitory activity of *Perilla frutescens* var. *crispa* leaves. Plants were cultivated under multiple fertilization conditions and harvested every 4 weeks. (**A**–**D**) Aβ aggregation inhibitory activity of leaves harvested in 2019. Activity in leaves of plants cultivated in soil with varying amounts of (**A**) slaked lime (SL), (**B**) shell powder (Sp), and shell granular (Sg) fertilizer (Table 1). (**C**,**D**) Activity in leaves of plants cultivated in soil with a combination of ammonium sulfate, lime superphosphate lime, and potassium sulfate fertilizer. (**E**) The activity of (**C**) was compared on September 5, just before flowering. Data represent the mean ± SD (*n* = 3, separate plants).

**Figure 7 foods-12-00486-f007:**
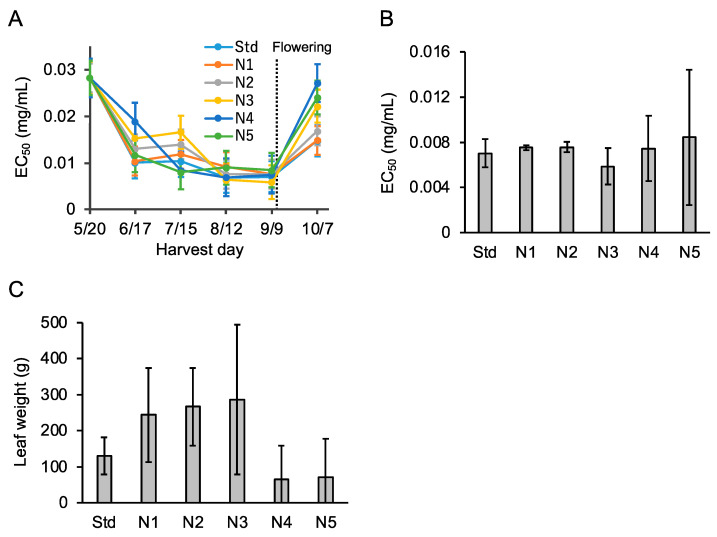
Nitrogen fertilization enhanced Aβ aggregation inhibitory activity and leaf weight. (**A**) Aβ aggregation inhibitory activity in leaves of *Perilla frutescens* var. *crispa* cultivated in a plastic greenhouse with six levels of ammonium sulfate fertilization (Table 2) and harvested every four weeks in 2020. (**B**,**C**) Activity (**B**) and leaf weight (**C**) just before flowering on September 9 in (**A**). Data represent as mean ± SD (*n* = 3, separate plants).

**Figure 8 foods-12-00486-f008:**
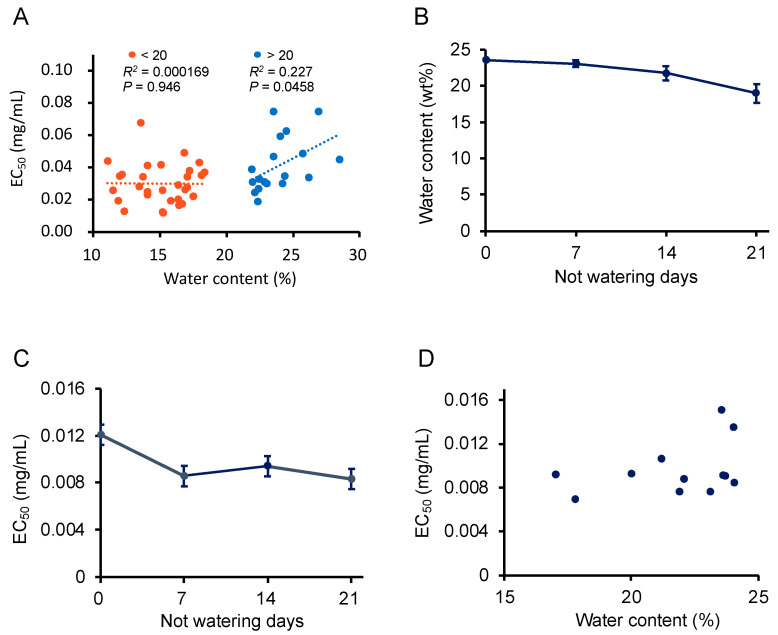
Drying enhanced the Aβ aggregation inhibitory activity of *Perilla frutescens* var. *crispa* leaves. (**A**) Scatter plot of soil water content at harvesting and EC_50_ of leaf extracts from *P*. *frutescens* var. *crispa* under conditions described in Table 1 harvested on 11 July, 8 August, and 5 September in 2019. Linear regression was performed on more and less than 20% s water content. (**B**–**D**) Plants were cultivated in greenhouses in 2020, and watering was stopped every week from the day before the harvest date (15 September) to determine their Aβ aggregation inhibitory activity (**C**) and soil water content (**B**). A scatter plot was created in (**D**) about water content and EC_50_ similar to (**A**). (**A**,**D**) represented as mean, and (**B**,**C**) represented as mean ± SD (*n* = 3, separate plants).

**Figure 9 foods-12-00486-f009:**
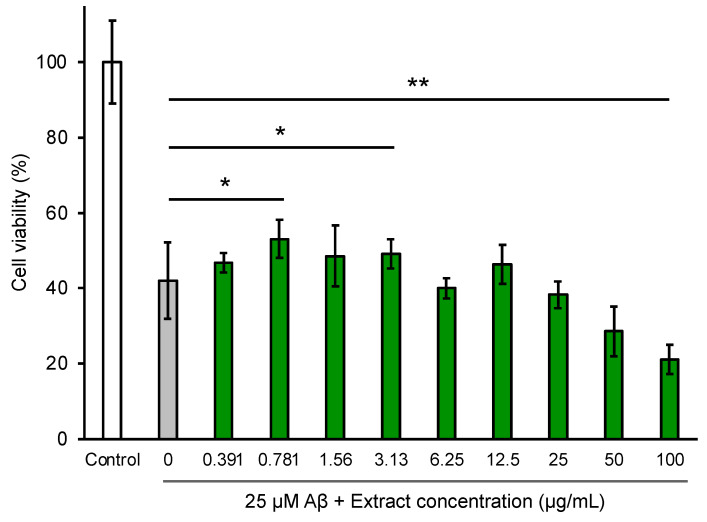
Extracts from *Perilla frutescens* var. *crispa* leaves suppressed Aβ-induced cytotoxicity in NGF-differentiated PC12 cells. PC12 cells were differentiated by NGF for 24 h and treated with 25 μM Aβ_42_ and leaf extract for 24 h. Cell viability was measured by the MTT assay and shown as the relative percentage of absorbance of treated samples compared to the control without Aβ and extract. Each plot and bar graph represent the mean ± SD (*n* = 3 separate experiments with extracts, 6 separate experiments in control, and 25 μM Aβ only). Comparison of the values (*: *p* < 0.05, **: *p* < 0.01, Welch’s *t*-test).

**Table 1 foods-12-00486-t001:** Multiple Fertilization Conditions in 2019.

Sample Name	Additional Fertilization	Amount (g/m^2^)
Std	-	
SL-1	Slaked lime	55.5
SL-2	Slaked lime	555.2
Cp	Calcium carbonate powder	163.5
Sp	Shell powder	170.3
Sg-1	Shell granular (particle size; 0.50–1.00 mm)	340.7
Sg-2	Shell granular (particle size; 1.00–2.00 mm)	340.7
Sg-3	Shell granular (particle size; 3.35–5.60 mm)	340.7
P	Superphosphate of lime (SPL)	114.3
N	Ammonium sulfate (AMS)	95.2
K	Potassium sulfate (SOP)	40.0
PN	Combination of P and N	
PK	Combination of P and K	
NK	Combination of N and K	
PNK	Combination of P, N, and K	
Sp-PNK	Combination of Sp, P, N, and K	

**Table 2 foods-12-00486-t002:** Fertilizer conditions of nitrogen in 2020.

Sample Name	Amount of Ammonium Sulphate (g/m^2^)
Std	0
N1	119
N2	238
N3	476
N4	952
N5	1905

## Data Availability

Data is available upon request from the corresponding authors.

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
