# Peer review of "Cultivation Factors That Affect Amyloid-β Aggregation Inhibitory Activity in Perilla frutescens var. crispa"

_foods, 2023, doi:10.3390/foods12030486_

Round 1

Reviewer 1 Report

The paper presents interesting study results based on influence of cultivation environment on anti-amyloid activity of P. frutescens var. crispa. On the whole the paper is well prepared.  The methods are adequate to aim of the studies. All methods are detail described. Statistical analysis is provided. Introduction include the most important information along with up-to-date references. Results are detailed described and discussed. Authors decided to use figures and charts in order to present the results. Please see the points below:

1.       Preparation of extracts: why scientists decided to various time of extraction (one day in 2019/2020 or seven days in 2018.)?

2.       Cultivation of perilla: please provide geographical coordinates for all regions of fields in Hokkaido

3.       Discussion: Authors should provide information about current new substances which are considered as potential anti-Ab drugs or are registered as drugs (i.e. aducanumab) and their activity should be compare to perilla activity.

Author Response

1; Preparation of extracts: why scientists decided to various time of extraction (one day in 2019/2020 or seven days in 2018.)?

We appreciate the opportunity to clarify this point. Until 2018, we performed extraction for seven days to extract more components from the perilla. In 2019/2020, we adopted one day of extraction time to prepare many samples. This is because we were able to confirm that the amyloid-β aggregation inhibitory activity did not change between one day and seven days of extraction time (Supplemental Figure S1).

We rewrote and added the sentence, “In 2019/2020, perilla leaf extracts were prepared by extraction for one day to prepare many samples efficiently.” into the Materials and Methods section (Revised manuscript, Page 4, lines 162 to 163).

We also added two sentences, “In 2019/2020, we prepared extracts efficiently with a change in extraction time from seven days to one day and evaluated many samples to investigate the details of cultivation conditions. We confirmed no change of Aβ aggregation inhibitory activity by extraction time (Supplemental Fig. S1).” into the Results section (Revised manuscript, Page 5, lines 209 to 212).

2; Cultivation of perilla: please provide geographical coordinates for all regions of fields in Hokkaido

We agree with your suggestion. According to your comment, we added the geographical coordinates of each field to the Materials and Methods section (Revised manuscript, Page 4, lines 153 to 155).

3; Discussion: Authors should provide information about current new substances which are considered as potential anti-Ab drugs or are registered as drugs (i.e. aducanumab) and their activity should be compare to perilla activity.

We are grateful for the exciting point out. While we agree with your interesting comment, we need to discuss the difference between Anti-Aβ antibody drugs (e.g., aducanumab and lecanemab) and perilla and other foods in approach to AD. Anti-Aβ antibody drugs selectively bind to Aβ aggregates in the brain and clear Aβ plaques by phagocytosis of microglia. In other words, these drugs are administered to AD patients to remove preformed aggregates of Aβ. In this study, we would like to emphasize the importance of preventing the development of AD, namely the progression of Aβ aggregation in the brain through daily diet. Here, we focused on the inhibition of Aβ aggregates formation using perilla extracts. According to your comment, to help readers understand, I explained the difference in the concept between Anti-Aβ antibody drugs and Foods-derived extraction samples in the introduction section and the discussion section. We added the sentences, “In addition to treatment by removing Aβ aggregation with anti-Aβ antibody drugs, we suggest the importance of adding a preventive effect against Aβ aggregation to the daily diet.” (Revised manuscript, Page 2, lines 60 to 62) and “In addition to treatment with anti-Aβ antibody drugs (e.g., aducanumab, lecanemab), prevention with foods such as Perilla may protect many people from AD risk.” (Revised manuscript, Page 14, lines 421 to 423).

Furthermore, we should discuss the effectiveness of perilla extracts. Therefore, we compared Aβ aggregation inhibitory activity perilla extracts and 504 natural plant extracts in Hokkaido, Japan we had already reported (Sasaki et al., 2019). In comparing Aβ aggregation inhibitory activity as an alternative, the highest activity of N3 (EC50 = 0.0059 mg/mL; Figure 7B) in this study was the second highest in the 504 natural plant extracts previously evaluated. Therefore, we added the sentences, “In particular, this study's highest activity of N3 (EC50 = 0.0059 mg/mL; Figure 7B) was the second highest in the 504 natural plant extracts previously evaluated [18].” into the Discussion section (Revised manuscript, Page 12, lines 345 to 347).

Reviewer 2 Report

In this study, the method of quantum-dot nanoprobes was used to investigate cultivation factors that affect Aβ aggregation inhibitory activity in Perilla frutescens var. cripa. It is of great significance for the rapid identification of active substances in plant foods and the search for medicine to prevent and treat Alzheimer's disease. There are several concerns as below.

1. Line114: mixed solutions were prepared with 6 concentrations of extracts. However, only 5 concentrations are shown in Figure 1B.

2. Line133:2 in CO2 should be subscript.

3. Line229-230: In particular, total leaf weight improved from Std to N3 but decreased significantly above N4. Although there is a difference between N3 and N4 in Figure 7C, there is no marked statistical difference and the word significantly is not appropriate.

Author Response

1; Line114: mixed solutions were prepared with 6 concentrations of extracts. However, only 5 concentrations are shown in Figure 1B.

Thank you for your accurate comment. We conducted an evaluation to prepare the mixed solution with 6 concentration extracts using automated microliter-scale high throughput screening. Actually, we eliminated the highest extract concentration (0.1 mg/mL) from the calculation because of the existence of insoluble substances affecting the standard deviation (SD) values. Therefore, we have added a sentence, “The images on the highest extract concentration, including insoluble substances, were eliminated because these affect SD values and disrupt accurate evaluation.” into the Materials and Methods section (Revised manuscript, Page 3, lines 123 to 125).

2; Line133:2 in CO2 should be subscript.

Thank you for pointing it out. As reviewer's comment, we corrected it to “CO2” (Revised manuscript, Page 3, line 138).

3; Line229-230: In particular, total leaf weight improved from Std to N3 but decreased significantly above N4. Although there is a difference between N3 and N4 in Figure 7C, there is no marked statistical difference and the word significantly is not appropriate.

We agree with your comment, “the word significantly is not appropriate”. According to the reviewer's comment, we deleted “significantly” from the sentence (Revised manuscript, Page 5, line 241).